# Impact of providing free HIV self-testing kits on frequency of testing among men who have sex with men and their sexual partners in China: A randomized controlled trial

Ci Zhang[1], Deborah Koniak-Griffin[2], Han-Zhu Qian[1,3], Lloyd A. Goldsamt[4], Honghong Wang[1], Mary-Lynn Brecht[2], Xianhong Li[1]*

**1** Xiangya School of Nursing, Central South University, Changsha, Hunan Province, China, **2** School of Nursing, University of California at Los Angeles, Los Angeles, California, United States of America, **3** School of Public Health, Yale University, New Haven, Connecticut, United States of America, **4** Rory Meyers College of Nursing, New York University, New York, New York, United States of America

* xianhong_li@csu.edu.cn

## Abstract

### Background

The HIV epidemic is rapidly growing among men who have sex with men (MSM) in China, yet HIV testing remains suboptimal. We aimed to determine the impact of HIV self-testing (HIVST) interventions on frequency of HIV testing among Chinese MSM and their sexual partners.

### Methods and findings

This randomized controlled trial was conducted in 4 cities in Hunan Province, China. Sexually active and HIV-negative MSM were recruited from communities and randomly assigned (1:1) to intervention or control arms. Participants in the control arm had access to site-based HIV testing (SBHT); those in the intervention arm were provided with 2 free finger-prick-based HIVST kits at enrollment and could receive 2 to 4 kits delivered through express mail every 3 months for 1 year in addition to SBHT. They were encouraged to distribute HIVST kits to their sexual partners. The primary outcome was the number of HIV tests taken by MSM participants, and the secondary outcome was the number of HIV tests taken by their sexual partners during 12 months of follow-up. The effect size for the primary and secondary outcomes was evaluated as the standardized mean difference (SMD) in testing frequency between intervention and control arms.

Between April 14, 2018, and June 30, 2018, 230 MSM were recruited. Mean age was 29 years; 77% attended college; 75% were single. The analysis population who completed at least one follow-up questionnaire included 110 (93%, 110/118) in the intervention and 106 (95%, 106/112) in the control arm. The average frequency of HIV tests per participant in the intervention arm (3.75) was higher than that in the control arm (1.80; SMD 1.26; 95% CI

**Data Availability Statement:** The study data are available upon request by contacting Ms. Yan Shen at yan_shen@csu.edu.cn, who is independent of the authors.

**Funding:** This study was supported by Fogarty International (DKG; D43 TW009579; https://www.nih.gov/), Central South University Innovation-driven project (XL; 2018CX036; http://www.csu.edu.cn/), and Central South University Graduate Independent Exploration Innovation project (CZ; 2018zzts885; http://www.csu.edu.cn/).The funders had no role in study design, data collection and analysis, decision to publish, or preparation of the manuscript.

**Competing interests:** The authors have declared that no competing interests exist.

**Abbreviations:** AOR, adjusted odds ratio; ARR, adjusted rate ratio; ART, antiretroviral therapy; CBO, community-based organizations; CDC, Centers for Disease Prevention and Control; CI, confidence interval; HIVST, HIV self-testing; MSM, men who have sex with men; PHT, partner HIV testing; PLWH, people living with HIV/AIDS; RCT, randomized controlled trial; SBHT, site-based HIV testing; SD, standard deviation; SMD, standardized mean difference; ZIP, Zero-inflated Poisson.

0.97–1.55; $P < 0.001$). This difference was mainly due to the difference in HIVST between the 2 arms (intervention 2.18 versus control 0.41; SMD 1.30; 95% CI 1.01–1.59; $P < 0.001$), whereas the average frequency of SBHT was comparable (1.57 versus 1.40, SMD 0.14; 95% CI −0.13 to 0.40; $P = 0.519$). The average frequency of HIV tests among sexual partners of each participant was higher in intervention than control arm (2.65 versus 1.31; SMD 0.64; 95% CI 0.36–0.92; $P < 0.001$), and this difference was also due to the difference in HIVST between the 2 arms (intervention 1.41 versus control 0.36; SMD 0.75; 95% CI 0.47–1.04; $P < 0.001$) but not SBHT (1.24 versus 0.96; SMD 0.23; 95% CI −0.05 to 0.50; $P = 0.055$). Zero-inflated Poisson regression analyses showed that the likelihood of taking HIV testing among intervention participants were 2.1 times greater than that of control participants (adjusted rate ratio [RR] 2.10; 95% CI 1.75–2.53, $P < 0.001$), and their sexual partners were 1.55 times more likely to take HIV tests in the intervention arm compared with the control arm (1.55, 1.23–1.95, $P < 0.001$). During the study period, 3 participants in the intervention arm and none in the control arm tested HIV positive, and 8 sexual partners of intervention arm participants also tested positive. No other adverse events were reported. Limitations in this study included the data on number of SBHT were solely based on self-report by the participants, but self-reported number of HIVST in the intervention arm was validated; the number of partner HIV testing was indirectly reported by participants because of difficulties in accessing each of their partners.

## Conclusions

In this study, we found that providing free HIVST kits significantly increased testing frequency among Chinese MSM and effectively enlarged HIV testing coverage by enhancing partner HIV testing through distribution of kits within their sexual networks.

## Trial registration

Chinese Clinical Trial Registry ChiCTR1800015584.

---

Author summary

### Why was this study done?

- The HIV epidemic is rapidly growing among men who have sex with men (MSM) in China, yet HIV testing remains suboptimal.

- Provision of HIV self-testing (HIVST) kits has been shown to be effective in promoting HIV testing among MSM in high-income countries, but there is a scarcity of evidence about the effects in low- and middle-income countries such as China.

- WHO guidelines advocate partner HIV testing (PHT) through sexual network distribution of HIVST kits; however, there is no solid evidence on the effectiveness of HIVST on improving PHT among MSM.

### What did the researchers do and find?

- We explored the impact of providing free HIVST kits on HIV testing frequency among Chinese MSM and their sexual partners.

- During the 12-month study period, the total frequency of HIV testing among MSM in the intervention arm (mean = 3.75) was higher than those in the control arm (mean = 1.80). The standardized mean difference (SMD) was 1.26 (95% CI 0.97–1.55).

- The total frequency of HIV testing among sexual partners of MSM was higher in the intervention than in the control arm (2.65 versus 1.31). The SMD was 0.64 (95% CI 0.36–0.92).

### What do these findings mean?

- Results from this randomized controlled trial support the effectiveness of providing free HIVST kits on improving HIV testing frequency among MSM in a low- and middle-income country such as China.

- Provision of HIVST kits could significantly enhance PHT through distribution of kits from MSM to their sexual partners.

- This effective intervention has the potential to be easily scaled up through the social and sexual networks of MSM in China.

## Introduction

Of 37.9 million people living with HIV/AIDS (PLWH) globally in 2018, about 79% knew their status [1], representing an 11% gap on average to the United Nations' first goal of the "90-90-90" targets: diagnosing 90% of all HIV-positive persons, providing antiretroviral therapy for 90% of those diagnosed, and achieving viral suppression for 90% of those treated by 2020 [2]. This gap is larger among men who have sex with men (MSM) and varies geographically; for example, about 1 in 6 MSM in the United States versus 1 in 3 in Kenya, Malawi, and South Africa, did not know their HIV status [3]. Testing for HIV is the gateway to treatment, care, and prevention, and achieving the first target is the foundation for reaching all targets [4].

One significant characteristic of the evolving HIV epidemic in China in recent years has been the rapid increase of transmission between MSM, which accounted for 3.4% of new cases in 2007 and 25.5% in 2017 [5]. An estimated 32% of PLWH in China do not know their serostatus [6], and about 50% of MSM have never been tested for HIV [7]. Both Chinese and European guidelines recommended that individuals who are at high risk for HIV infection should be tested every 3 to 6 months [8,9], and some novel approaches have been examined for increasing uptake of HIV testing among MSM [10–12]; however, 55% of Chinese MSM at high risk did not meet this suggested testing frequency [13].

Delayed HIV testing and antiretroviral therapy (ART) increase the probability of death due to AIDS-related illnesses [14]. PLWH who do not know their HIV status are more likely to transmit HIV than those who do [15]. Recent longitudinal studies showed that with timely diagnosis and effective suppression of HIV viral load, the risk of transmission among serodiscordant gay couples through condomless intercourse was zero [16–18].

There are a variety of barriers for MSM to take an HIV test at hospitals or Centers for Disease Prevention and Control (CDC) in China [19]. Compared with site-based HIV testing (SBHT) in healthcare facilities, CDC clinics, and community-based organizations (CBO), HIV self-testing (HIVST) has advantages in privacy, confidentiality, and convenience [20], and therefore has the potential to overcome the barriers for access to SBHT services, such as HIV/gay-related stigma and discrimination, long waiting time for testing and/or receipt of results at the sites, and lengthy commutes to the sites [19]. Findings from studies conducted in the US and Australia suggested that HIVST improved HIV testing frequency for high-risk MSM [21–23]. To our knowledge, there has only been one study conducted in China (in Hong Kong) to explore the impact of HIVST on HIV testing coverage among MSM [24]. WHO recommends HIVST as a complementary approach to SBHT services to improve HIV testing coverage and frequency [25]. However, there is a scarcity of research to explore the impact of HIV testing frequency among MSM conducted in China, where HIVST kits have not been approved for personal use, although several types of rapid HIV testing kits are available for professional use [26].

WHO encourages expanding partner HIV testing (PHT) [25,27], which was defined as distribution of HIVST kits to sexual partners or recommending sexual partners to test for HIV at SBHT organizations. Previous research indicated that provision of free HIVST kits to female sexual workers or women receiving antenatal and postpartum care could increase their male sexual partners' HIV testing uptake [28]. MSM are known to have strong sexual networks, and evidence showed that the proportion of HIV-positive MSM infected by casual or regular partners has been increasing during recent decades [29]. Thus, promoting PHT among MSM by providing HIVST kits may have significant implications. However, although literature has indicated that MSM have a high acceptability and willingness (79%–91%) to deliver kits to their partners [30], the effectiveness of this approach has not been rigorously examined.

We conducted a randomized controlled trial (RCT) to test the hypothesis that provision of free HIVST kits to MSM could increase their HIV testing frequency and enlarge HIV testing coverage by enhancing PHT through the distribution of HIVST kits within MSM sexual networks.

## Methods

### Formative research

During development of the study protocol, we conducted formative research by surveying 23 MSM. Survey questions included acceptability of HIVST, preference for oral or finger-stick self-testing kits, preference for the way of delivering testing kits, and perception on partner testing. The majority (87%, 20/23) reported willingness to use HIVST, 89% (17/19) preferred finger-stick rather than oral rapid self-testing kits, all chose using express mail for delivering HIVST kits, and 87% (13/15) said it feasible to do partner testing [31]. Findings from this formative research were used to guide protocol development. Men in this survey were not involved in the study design, participant recruitment, or implementation of RCT.

### Study design and participants

This study was designed as a randomized trial to evaluate the impact of providing free HIVST kits on HIV testing coverage and frequency among MSM and their sexual partners in 4 cities of Hunan Province in central China, whose protocol was published elsewhere [31]. This study was performed following the Consolidated Standards of Reporting Trials (S1 CONSORT 2010 Checklist). These cities included the province's capital city Changsha and 3 prefecture-level cities—Changde, Shaoyang, and Yiyang—where there were no free HIVST kits provided by

either public health or other research programs during the study period, although people could buy rapid HIV testing kits online.

Through collaborating with Zuo An Cai Hong, a gay-friendly CBO located in Changsha with satellite offices in the other 3 study cities, we recruited MSM participants via community outreach, social media, and hotlines. Eligibility criteria included (1) born as a male; (2) aged 18 years or above; (3) self-reported condomless anal or oral sex with men in the past 3 months; (4) HIV seronegative by rapid screening testing at eligibility screening; (5) no plan to leave Hunan Province in the next 12 months; (6) possessing a smartphone and adept in using the social media app WeChat, which is universally used by urban adult residents; and (7) voluntarily agreeing to participate in the study and provide informed consent. Men were excluded if they scored above 35 on the Brief Psychiatric Rating Scale, which indicates having a psychiatric disorder [32], or if they could not read or speak Chinese.

## Randomization and masking

In order to balance the history of HIV testing among participants in 2 arms, we stratified eligible participants into 2 subgroups at enrollment: (1) recent testers who had at least 1 HIV test in the past 2 years; and (2) non-recent testers who had not tested for HIV in the past 2 years, including those who had never had an HIV test in their lifetime. Two separate randomized number tables for recent testers and non-recent testers were generated with SPSS V.18.0 (IBM SPSS Statistics, https://www.ibm.com/products/spss-statistics)) and used to divide participants randomly (at 1:1 allocation ratio) into the intervention and control arms.

Participant enrollment and randomization were conducted sequentially in 4 study cities. A statistician randomly generated treatment allocation for 2 participants at a time, one labeled as intervention and the other labeled as control; these labels were put in sealed opaque envelopes, which were then ordered in sequence. In each study city, once a participant had consented to enter a trial, an envelope was picked in sequence and opened, and the participant was then offered the allocated intervention or control. The research assistant enrolling the participants did not know the randomization outcome until the envelope was opened by participant. This randomization procedure was conducted in a private room, so next participant did not know the assignment of previous participant and therefore could not guess his own assignment. Randomized assignment was done separately for each subgroup.

## Procedures

Participants in the control condition were referred to local hospitals, free HIV testing and counseling clinics at local CDC, and gay-friendly CBOs to receive SBHT services. They could also buy HIVST kits online or from local pharmacies. Participants in the intervention condition were provided with free HIVST kits during 12 months of study follow-up in addition to receiving SBHT services at local hospitals, CDC clinics, and CBOs, and they were also encouraged to distribute kits to their sexual partners. Two HIVST kits were provided to each participant at enrollment, along with detailed electronic users' instructions and counseling information, including 24/7 hotlines and an official WeChat study account to reach research assistants to obtain consultation on the HIVST administration and interpretation of testing results (https://mp.weixin.qq.com/s/vi9Nl-uLOeWnoJajvsq63Q). Participants sent an electronic photocopy of their test result via secured individual WeChat contact with a research assistant (RA) after they used each HIVST kit, and then they were replenished with new free kits. Participants were eligible to receive 2 to 4 kits every 3 months, with a maximum of 12 kits for 1 year. This distribution plan was based upon research findings about a mean number of 4.79 sexual partners in the past 6 months [33] and recommendations for HIV testing

frequency of once every 3 to 6 months for high-risk MSM [8,9]. HIVST test kits were sent to the addresses the participants provided via express mail, or participants came to pick up kits at study clinics. MSM could also purchase HIVST kits online if they needed more than 4 kits every 3 months.

The finger-prick-based HIVST kit is a third generation of Alere Determine HIV 1/2 rapid assay (Alere Medical Co., Japan), which was approved by China's State Food and Drug Administration and the US Food and Drug Administration [26]. The sensitivity and specificity of this third generation were 100% and 99.7%, respectively [34]. HIV 1/2 antibodies in blood could be detected using this kit at 6 weeks after infection [34]. Participants and their sexual partners who had positive HIVST results were advised to contact the trained RA, who provided post-test counseling and referral to local CDCs for a confirmatory test.

## Outcomes

The primary outcome was the total number of HIV tests (HIVST plus SBHT) for MSM participants during 12 months of follow-up (HIV testing frequency). The secondary outcome was the total number of HIV tests for sexual partners of MSM participants during 12 months of follow-up (HIV testing coverage), which included HIVST that the participants provided and SBHT that the participants successfully recommended and was indirectly reported by MSM participants. Both were based on self-report of MSM participants. The self-reported number of HIVST in the intervention arm was validated by counting the number of HIVST kits provided by the study. Regular HIV testing was measured as proportions of "at least 1 HIV test" and "at least 2 HIV tests" during 12 months of follow-up.

Data were collected using questionnaires on an online platform (http://www.sojump.com) at baseline, 3-, 6-, 9-, and 12-month follow-ups. The baseline questionnaire assessed sociodemographic characteristics, number of male sexual partners, frequency of HIV testing among MSM participants including HIVST and SBHT in the past 12 months, and frequency of HIV testing among sexual partners including HIVST and SBHT in the past 12 months as reported by MSM participants. The follow-up questionnaires covered the frequency, results, and location of HIV testing (HIVST plus SBHT) during the 3-month follow-up intervals among MSM participants and their sexual partners. Participants and their sexual partners with a positive HIVST result were also asked whether they had HIV confirmatory laboratory tests and whether they had initiated HIV ART. Participants received 100 Chinese yuan (about 15 US dollars) as compensation for their time spent completing each round of the questionnaire, and the maximum amount of compensation for completing all study procedures was 500 yuan or 75 dollars. Participants received phone or WeChat text message reminders for completing follow-up questionnaires.

## Statistical analysis

Sample sizes were calculated for recent and non-recent testers separately based on normal distribution of HIV testing frequencies. G-power V.3.0 (Franz Faul, http://downloads.fyxm.net/G*Power-10787.html) was used to calculate the sample size considering a 5% significance level [35], 80% power, and a 1-tailed test. Recommendation from guidelines suggested 2 to 4 HIV tests per year for individuals at high risk [8,9]. Given the missing rate of 20%, we conservatively estimated 184 recent testers were needed to detect the increase from 1.35 to 2 (standard deviation [SD] 1.61) HIV tests per year [13], and 26 non-recent testers were needed to detect the increase from 0.2 to 1 (SD 0.7) HIV test per year [22]. The sample size of non-recent testers was small and did not support multivariate analysis. As non-recent testers were equally distributed in intervention and control arms, we combined non-recent testers and recent testers in

all data analyses. The subgroup variable was included as a confounding factor in all multivariate models.

Data were analyzed by SPSS V.18.0, exported directly from the *sojump* survey platform to avoid data entry errors. Data analysis included participants who completed at least 1 follow-up questionnaire. The proportion of missing data in all 4 follow-up questionnaire surveys was 4.4% (38/864). Considering it is less than 5%, we assumed data missed completely at random and filled the number of HIV tests through multiple imputations using R software V. 3.6 (R Foundation, https://www.r-project.org/) [36]. So, the number of HIV tests is the average frequency of tests taken by each participant during all follow-up surveys within 12 months of follow-up.

To evaluate the impact of intervention on HIV testing frequency among MSM participants (primary outcome) and their sexual partners (secondary outcome), we compared the outcome variables between intervention and control arms. For continuous variables with a normal distribution, independent sample *t*-tests were used to compare 2 arms; otherwise, rank-sum tests were used. The effect size for the primary and secondary outcomes was evaluated as the standardized mean difference (SMD), which was obtained by dividing the mean difference of testing frequency between intervention and control arms by their pooled SD, and 95% confidence interval (CI) for SMD was also calculated.

We also evaluated the regular HIV testing frequency by recoding the number of tests into 2 binary variables: $\geq 1$ or $< 1$ test, or $\geq 2$ or $< 2$ tests. Logistic regression was performed to examine the impact of intervention while adjusting for potential confounders, including subgroup, age, ethnicity, residence, cohabitation, duration of living in a study city, education level, sexual orientation, marital status, occupation, and monthly personal income. Adjusted odds ratios (AORs) were calculated, with 95% CIs. A significance level was set as a 2-sided *P* value less than 0.05.

Our data on HIV test frequency exhibit a greater proportion of zero counts than is consistent with the data having been generated by a simple Poisson or negative binomial process. To avoid a model misspecification that can result in biased or inconsistent estimators, the excess zeros should be accounted properly. Zero-inflated Poisson (ZIP) regression analysis was conducted to evaluate the impact of providing HIVST kits on HIV testing coverage and frequency among MSM participants and their partners by adjusting for potential confounders, including subgroup, age, ethnicity, residence, cohabitation, duration of living in a study city, education level, sexual orientation, marital status, occupation, and monthly personal income. Based on Chi-Square test, overdispersion was not detected.

## Ethics and dissemination

Oral consent for undertaking a baseline rapid HIV screening test and written informed consent prior to enrollment were obtained from each participant. Ethical approval of the study protocol was obtained from the Institutional Review Board for Behavioral and Nursing Research in Central South University Xiangya Nursing School (approval #2018002). The participants' personal information and study case record forms were stored in an offline encrypted computer, which was accessed by the principal investigator and her designated research team members. The results will be disseminated through academic journals and conferences after the study concluded.

## Trial registration

The study protocol was registered with Chinese Clinical Trial Registry (www.chictr.org.cn, ID: ChiCTR1800015584).

## Results

Between April 14, 2018, and June 30, 2018, a total of 230 MSM were enrolled in the study, including 134 in Changsha, 38 in Changde, 30 in Shaoyang, and 28 in Yiyang city; of these, 216 (93.9%) completed at least 1 follow-up questionnaire and were included in the data analysis. These included 190 recent and 26 non-recent testers, and 110 were in the intervention and 106 in the control arm (Fig 1).

Table 1 shows demographic characteristics, baseline sexual behavior, and HIV testing frequency. The mean age was 29 years; 77% attended college; 75% were single; 15% were students; and 72% considered themselves homosexual, 26% bisexual, and 2% heterosexual. In the 12 months prior to enrollment, the average number of sexual partners per participant was 5; the average frequency of HIV tests among MSM participants was 1.26, including 0.52 HIVST and 0.74 SBHT; and the average frequency of HIV testing among sexual partners was 0.77, including 0.27 HIVST and 0.50 SBHT.

During 12-month follow-up, the average frequency of HIV tests among MSM participants in the intervention arm (mean 3.75) was higher than HIV testing among MSM in the control arm (1.80; SMD 1.26; 95% CI 0.97–1.55; $P < 0.001$). This difference was mainly due to HIVST

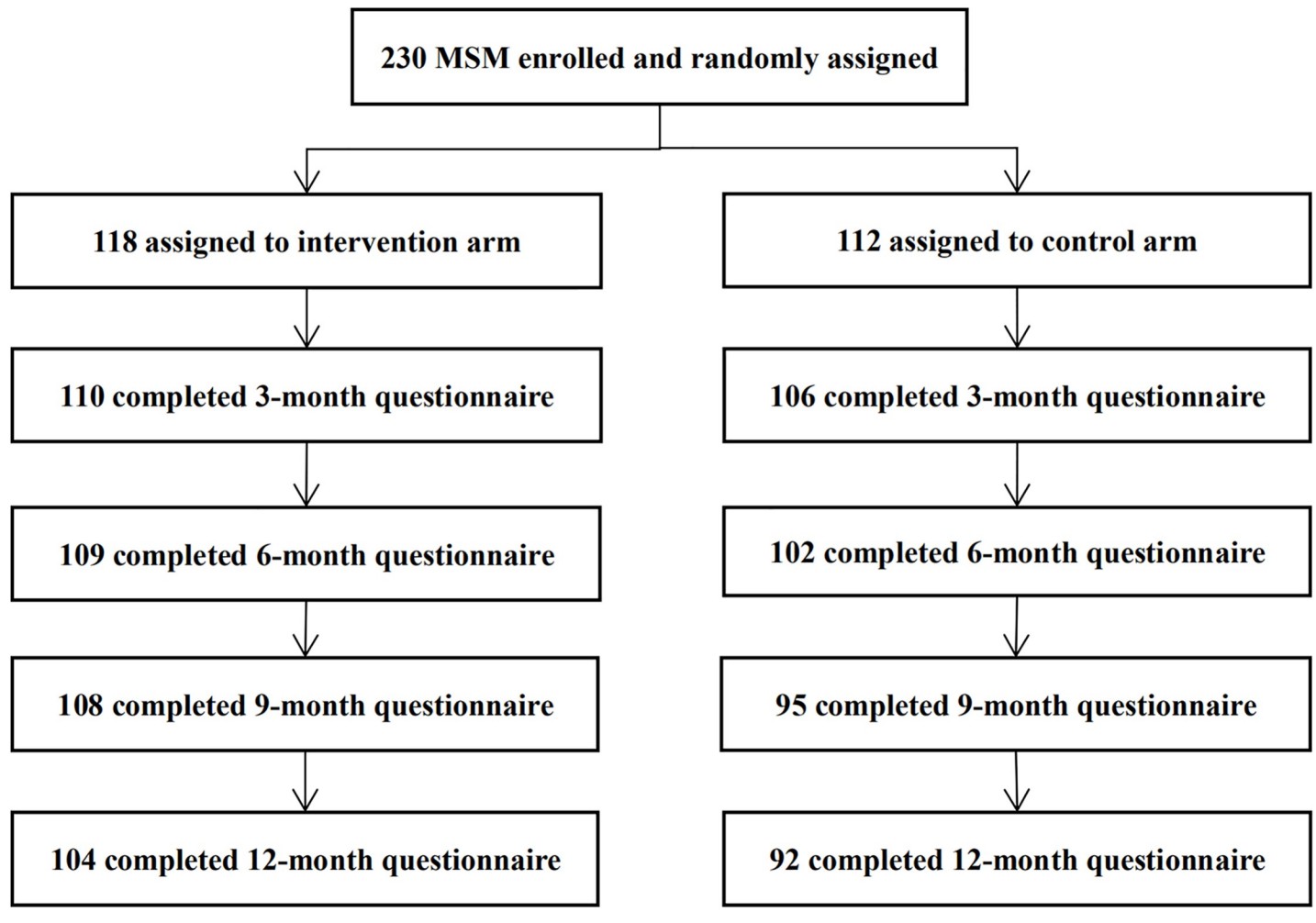

**Fig 1. Trial profile.** MSM, men who have sex with men.

**Table 1. Demographic characteristics and baseline sexual behaviors and HIV testing among MSM participants.**

| | Total (N = 216) | Intervention (N = 110) | Control (N = 106) |
|---|---|---|---|
| **Age, year, mean (SD)** | | | |
| | 29.0 (7.7) | 29.2 (7.8) | 28.5 (8.0) |
| **History of HIV testing in the 24 months prior to enrollment** | | | |
| Recent testers who had a test | 190 (88.0) | 96 (87.3) | 94 (88.7) |
| Non-recent testers who did not have a test | 26 (12.0) | 14 (12.7) | 12 (11.3) |
| **Ethnicity** | | | |
| Han majority | 202 (93.5) | 101 (95.3) | 101 (91.8) |
| Minority | 14 (6.5) | 5 (4.7) | 9 (8.2) |
| **Residence** | | | |
| Urban | 149 (69.0) | 78 (70.9) | 71 (67.0) |
| Rural | 67 (31.0) | 32 (29.1) | 35 (33.0) |
| **Cohabitation** | | | |
| Living alone | 51 (23.6) | 22 (20.0) | 29 (27.4) |
| Living with family | 97 (44.9) | 57 (51.8) | 40 (37.7) |
| Living with others | 68 (31.5) | 31 (28.2) | 37 (34.9) |
| **Duration of living in a study city** | | | |
| <2 years | 47 (21.8) | 20 (18.2) | 27 (25.5) |
| ≥ 2 years | 169 (78.2) | 90 (81.8) | 79 (74.5) |
| **Education level** | | | |
| Ever attended college | 163 (75.5) | 86 (78.2) | 77 (72.6) |
| No college education | 53 (24.5) | 24 (21.8) | 29 (27.4) |
| **Sexual orientation** | | | |
| Homosexual | 155 (71.8) | 81 (73.6) | 81 (73.6) |
| Bisexual | 57 (26.4) | 27 (24.5) | 27 (24.5) |
| Heterosexual | 4 (1.9) | 2 (1.8) | 2 (1.8) |
| **Marital status** | | | |
| Never married | 165 (76.4) | 86 (78.2) | 79 (74.5) |
| Ever married | 51 (23.6) | 24 (21.8) | 27 (25.5) |
| **Occupation** | | | |
| Students | 40 (18.5) | 24 (21.8) | 16 (15.1) |
| Nonstudents | 176 (81.5) | 86 (78.2) | 90 (84.9) |
| **Monthly personal income** | | | |
| <2,000 yuan (approximately 300 USD) | 47 (21.8) | 20 (18.2) | 27 (25.5) |
| ≥2,000 yuan (approximately 300 USD) | 169 (28.2) | 90 (81.8) | 79 (74.5) |
| **Number of sexual partners in past 12 months, mean (SD)** | | | |
| | 5.04 (2.21) | 5.15 (2.25) | 4.92 (2.17) |
| **Frequency of HIV tests among MSM participants in the past 12 months, mean (SD)** | | | |
| HIVST | 0.52 (0.86) | 0.54 (0.84) | 0.51 (0.89) |
| SBHT | 0.74 (1.11) | 0.75 (1.22) | 0.73 (1.00) |
| HIVST+SBHT | 1.26 (1.23) | 1.28 (1.31) | 1.24 (1.52) |
| **Frequency of HIV tests among sexual partners of MSM participant in the past 12 months, mean (SD)** | | | |
| HIVST | 0.27 (0.75) | 0.27 (0.69) | 0.27 (0.81) |
| SBHT | 0.50 (0.99) | 0.47 (0.98) | 0.52 (1.00) |
| HIVST+SBHT | 0.77 (1.37) | 0.75 (1.29) | 0.79 (1.46) |

Data in the table are n (%), except where noted as mean (SD).

HIVST, HIV self-testing; MSM, men who have sex with men; PHT, partner HIV testing; SBHT, site-based HIV testing; SD, standard deviation; USD, US dollars.

**Table 2. Impact of providing self-test kits on HIV testing coverage and frequency among MSM participants during 12 months of follow-up.**

| | Total (n = 216) | Intervention (n = 110) | Control (n = 106) | OR (95% CI) | Adjusted OR (95% CI) | SMD (95% CI) | Z or t | P |
|---|---|---|---|---|---|---|---|---|
| **Frequency of HIV tests per participant (mean, SD)** | 2.78 (1.81) | 3.75 (1.77) | 1.80 (1.28) | - | - | 1.26 (0.97–1.55) | −7.64 | <0.001 |
| **≥ 1 test (n, %)** | 196 (90.7) | 104 (94.5) | 92 (86.6) | 2.64 (0.97–7.15) | 4.32 (1.38–13.56)* | - | - | 0.012 |
| **≥ 2 tests (n, %)** | 153 (70.8) | 93 (84.5) | 60 (56.6) | 4.19 (2.20–7.99) | 5.09 (2.56–10.11)* | - | - | <0.001 |
| **Frequency of HIVST (mean, SD)** | 1.31 (1.63) | 2.18 (1.77) | 0.41 (0.75) | - | - | 1.30 (1.01–1.59) | -8.58 | <0.001 |
| **Frequency of SBHT (mean, SD)** | 1.49 (1.25) | 1.57 (1.35) | 1.40 (1.13) | - | - | 0.14 (−0.13 to 0.40) | -0.66 | 0.519 |

*Adjusted OR: Adjusted for subgroup, age, ethnicity, residence, cohabitation, duration of living in a study city, education level, sexual orientation, marital status, occupation, and monthly income.

CI, confidence interval; HIVST, HIV self-testing; MSM, men who have sex with men; OR, odds ratio; SBHT, site-based HIV testing; SD, standard deviation; SMD, standardized mean difference.

between 2 arms (intervention 2.18 versus control 0.41; SMD 1.30; 95% CI 1.01–1.59; $P < 0.001$), whereas the average frequency of SVHT was comparable (1.57 versus 1.40; SMD 0.14; 95% CI −0.13 to 0.40; $P = 0.519$) (Table 2).

ZIP regression analysis showed that after adjustment for subgroup and demographic variables, the likelihood of taking HIV testing among intervention participants was 2.1 times greater than control participants (adjusted rate ratio [ARR] 2.10, 95% CI 1.75–2.53, $P < 0.001$). Subgroup, age, ethnicity, residence, duration of living in a study city, cohabitation, education, sexual orientation, marriage status, being a student or not, and income are not significantly associated with the number of HIV tests among MSM participants (Table 3).

A higher proportion of participants in the intervention arm performed at least 1 HIV test during 12 months of follow-up than in the control arm (95% versus 87%, $P = 0.021$). Participants in the intervention arm were also more likely to perform at least 2 tests than those in the control arm (85% versus 57%, $P < 0.001$), which reflects the suggested testing guideline that high-risk MSM should take a test every 3 to 6 months. Intervention participants were 4 times more likely to take at least 1 test during follow-up than control participants (AOR 4.32; 95% CI 1.38–13.56), and they were 5 times more likely to take at least 2 tests than control participants (5.09, 2.56–10.11) (Table 2).

Participants in the intervention arm reported a higher frequency of HIV testing during the follow-up than those in the control arm; nearly 60% of intervention participants reported 4 or more tests in comparison to less than 10% of control participants (Fig 2).

There was no statistically significant difference in the mean number of sexual partners reported by MSM participants between the intervention and control arms (7.25 versus 6.98, $P = 0.912$). In total, the intervention arm reported 291 PHTs, whereas the control arm reported 138 PHTs. The average frequency of HIV tests among sexual partners of each participant was higher in intervention than control arm (mean 2.65 versus 1.31; SMD 0.64; 95% CI 0.36–0.92; $P < 0.001$), and this difference was also due to the difference in HIVST between 2 arms (intervention: 1.41 versus control: 0.36; SMD 0.75; 95% CI 0.47–1.04; $P < 0.001$) but not SBHT (1.24 versus 0.96; SMD 0.23; 95% CI −0.05 to 0.50; $P = 0.055$) (Table 4). ZIP regression analysis showed that the sexual partners of intervention participants were 1.55 times more likely to take HIV testing compared with the partners of control participants (ARR 1.55, 95% CI 1.23–1.95, $P < 0.001$) (Table 4).

**Table 3. Zero-inflated Poisson regression analysis of the impacts of providing self-test kits on HIV testing coverage and frequency among MSM participants during 12 months of follow-up*.**

| | Value estimates (SE, 95% CI) | Rate ratio/Adjusted rate ratio (95% CI) | P |
|---|---|---|---|
| **Study arm** Intervention versus Control | 0.69 (0.09, 0.51–0.87) | 1.99 (1.66–2.39) | <0.001 |
| **Adjusted by confounders**** | 0.74 (0.09, 0.56–0.93) | 2.10 (1.75–2.53) | <0.001 |
| **Subgroup** Recent testers versus Non-recent testers | −0.03 (0.15, −0.32 to 0.27) | 0.97 (0.73–1.31) | 0.852 |
| **Age** | 0.008 (0.005, −0.003 to 0.018) | 1.01 (0.99–1.02) | 0.153 |
| **Ethnicity** (Minorities versus Han majority) | 0.12 (0.15, −0.18 to 0.42) | 1.13 (0.83–1.52) | 0.418 |
| **Residence** (Urban versus Rural) | 0.03 (0.092, −0.15 to 0.21) | 1.03 (0.86–1.23) | 0.743 |
| **Duration** (<2 versus ≥2 yrs) | 0.11 (0.09, −0.08 to 0.30) | 1.12 (0.92–1.35) | 0.255 |
| **Cohabitation** | | | |
| Others versus Alone | −0.07 (0.11, −0.29 to 0.15) | 0.93 (0.75–1.16) | 0.535 |
| Family versus Alone | 0.01 (0.10, −0.19 to 0.21) | 1.01 (0.83–1.23) | 0.925 |
| **Education** (College versus No) | 0.08 (0.10, −0.11 to 0.27) | 1.08 (0.90–1.31) | 0.436 |
| **Orientation** (Homosexual versus Other) | 0.05 (0.03, −0.13 to 0.23) | 1.05 (0.88–1.26) | 0.616 |
| **Marriage** (Married versus No) | 0.09 (0.09, −0.09 to 0.28) | 1.09 (0.91–1.32) | 0.330 |
| **Occupation** (Students versus No) | 0.02 (0.11, −0.19 to 0.23) | 1.02 (0.83–1.26) | 0.829 |
| **Income** (<2000 versus ≥ 2000 Yuan) | −0.03 (0.11, −0.24 to 0.18) | 0.97 (0.79–1.20) | 0.779 |

*The frequency of HIV tests for sexual partners of MSM was used as linear predictor in each of the following models

**Logit model was used to model the probability of a zero count. ^Adjusted by 11 variables: subgroup, age, ethnicity, residence, duration, cohabitation, education, orientation, marriage, occupation, and monthly income.

CI, confidence interval; MSM, men who have sex with men; SE, standard error.

The sexual partners of intervention participants had a higher proportion of at least 1 test than those in the control participants (79% versus 55%; $P < 0.001$), and the proportion was more than 3 times higher in the former than in the latter arm (AOR, 4.02, 95% CI 2.05–7.86). Sexual partners of intervention participants were also more likely to take at least 2 tests than partners of control participants (59% versus 40%, $P < 0.001$), and the odds in the intervention arm were double those in the control arm (2.85, 1.57–5.18) (Table 5).

During the 12 months of follow-up, 3 MSM participants (incident cases) in the intervention arm and none in the control arm tested HIV positive, and 8 sexual partners of intervention arm participants also tested positive (incident or prevalent cases). All HIV-positive individuals had laboratory HIV confirmation tests and were linked to care. No adverse events (including disclosure of homosexuality, related psychological and social impacts such as discrimination, loss of job, or suicide and so on) were reported. Less than 20% of participants in the intervention arm, as compared with nearly 50% in the control arm, reported that none of their sexual partners took a test (Fig 2).

## Discussion

### Main findings

This RCT demonstrated that providing free HIVST kits significantly increased testing frequency for Chinese MSM and effectively enlarged HIV testing coverage by enhancing PHT through the distribution of HIVST kits within their sexual networks. The intervention effect satisfied the requirement for HIV testing frequency recommended by China CDC and has more than 2 times greater capacity to enlarge HIV testing coverage than the SBHT approach.

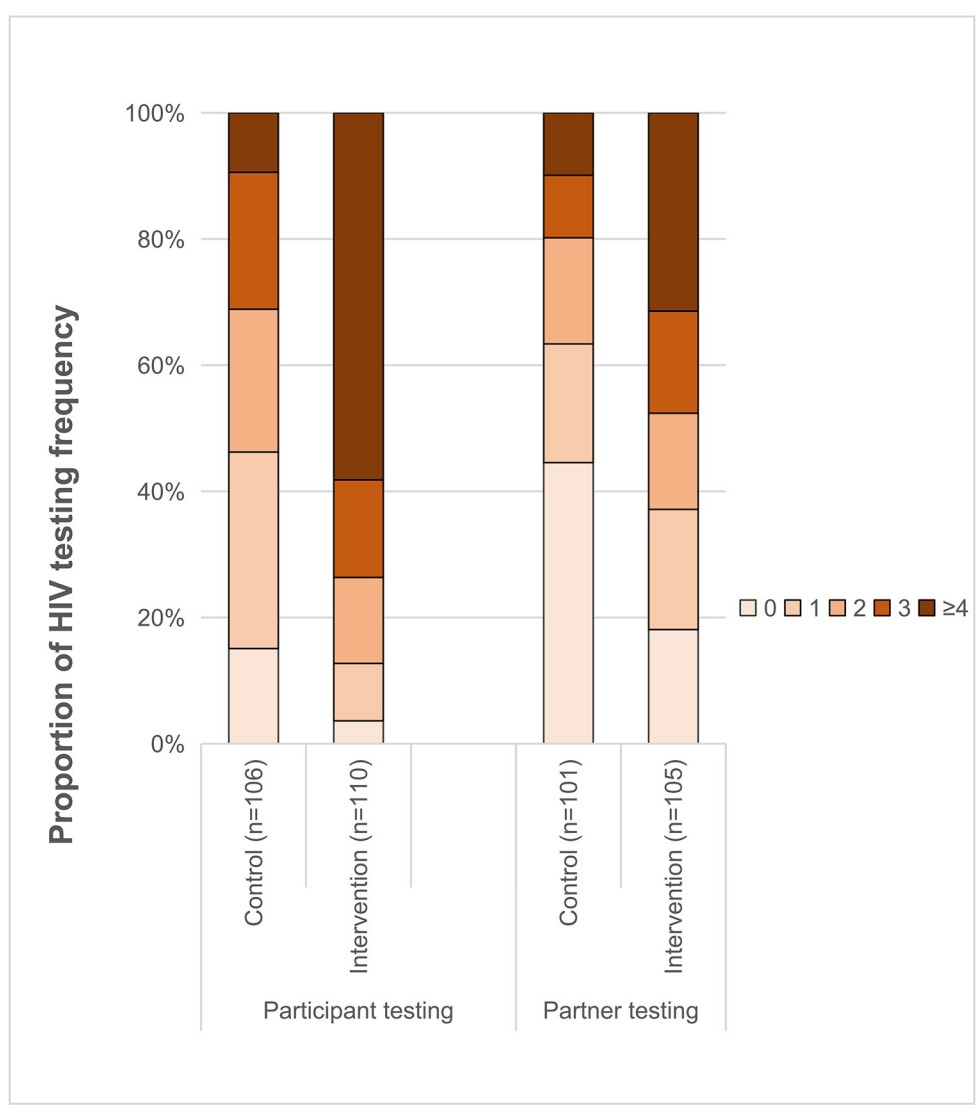

**Fig 2. Proportion of HIV testing frequency.**

**Table 4. Zero-inflated Poisson regression analysis of the impacts of providing self-test kits on HIV testing frequency among partners of MSM participants during 12 months of follow-up***.

| | Value Estimates (SE, 95% CI) | Rate ratio/Adjusted rate ratio (95% CI) | *P* |
|---|---|---|---|
| **Study arm** Intervention versus control | 0.41 (0.11, 0.18–0.63) | 1.51 (1.20–1.88) | <0.001 |
| **Adjusted by confounders*** | 0.44 (0.12, 0.21–0.67) | 1.55 (1.23–1.95) | <0.001 |

*The frequency of HIV tests among MSM was used as linear predictor in each of following models

**Logit model was used to model the probability of a zero count. ^Adjusted by 11 variables: subgroup, age, ethnicity, residence, duration, cohabitation, education, orientation, marriage, occupation, and monthly income.

CI, confidence interval; MSM, men who have sex with men; SE, standard error.

**Table 5. Impact of providing HIV self-test kits on testing coverage and frequency among sexual partners of MSM participants during 12 months of follow-up.**

| | Total (n = 207*) | Intervention (n = 105) | Control (n = 102) | OR (95% CI) | Adjusted OR (95% CI) | SMD (95% CI) | Z or t | P |
|---|---|---|---|---|---|---|---|---|
| Frequency of HIV tests among sexual partners of each MSM participant (mean, SD) | 2.00 (2.20) | 2.65 (2.45) | 1.31 (1.65) | - | - | 0.64 (0.36–0.92) | −4.54 | <0.001 |
| ≥ 1 test (n, %) | 145 (67.1) | 87 (79.1) | 58 (54.8) | 3.13 (1.72–5.69) | 4.02 (2.05–7.86)* | - | - | <0.001 |
| ≥ 2 tests (n, %) | 104 (48.1) | 65 (59.1) | 39 (36.8) | 2.48 (1.44–4.29) | 2.85 (1.57–5.18)* | - | - | <0.001 |
| Frequency of HIVST (mean, SD) | 0.89 (1.49) | 1.41 (1.74) | 0.36 (0.90) | - | - | 0.75 (0.47–1.04) | −5.99 | <0.001 |
| Frequency of SBHT (mean, SD) | 1.10 (1.25) | 1.24 (1.31) | 0.96 (1.16) | - | - | 0.23 (−0.05 to 0.50) | −1.56 | 0.055 |

*Adjusted OR: Adjusted for subgroup, age, ethnicity, residence, cohabitation, duration of living in a study city, education level, sexual orientation, marital status, occupation, and monthly income.

CI, confidence interval; HIVST, HIV self-testing; MSM, men who have sex with men; OR, odds ratio; SBHT, site-based HIV testing; SD, standard deviation; SMD, standardized mean difference.

Note: Nine MSM participants did not have sexual partners during 12 months of follow-up.

This effective intervention has the potential to be easily scaled up through the social and sexual networks of MSM in China.

## Strengths and limitations

This study has several strengths. First, to our knowledge, it is the first RCT to examine whether providing HIVST kits increases the frequency of HIV testing among a high-risk population of MSM in China, a low- and middle-income country. The results show that the average number of HIV tests among MSM was 3.75 after adding HIVST as a complementary testing approach, which satisfied the guidelines on HIV testing frequency [8,9]. Second, this is the first RCT, to our knowledge, that examines whether provision of HIVST kits can increase the HIV testing coverage and the frequency of PHT through the MSM's distribution of kits to their sexual partners. Prior to this study, there were sparse data assessing the effects of HIVST on PHT, despite the WHO recommendations on PHT [25,27]. Third, this RCT recruited participants by applying diverse methods, including in HIV testing sites, via an online platform, and from CBO referrals, which guaranteed the diversity of participants.

There are limitations in our study. First, the data on SBHT frequency were solely based on self-report by the participants. However, self-reported HIVST frequency in the intervention arm was compared with that in the control arm by counting the number of testing kits provided by the study, and the data were consistent. This provides indirect evidence on the validity of SBHT data. Second, the number of PHT was indirectly reported by participants because of difficulties in accessing each of their partners. This information bias may lead to underreporting of PHT frequency in both arms, but it may not change the study result showing a significant difference between 2 arms. Third, the requirement for a rapid HIV test for participant enrollment might exclude some MSM from this study who did not want to take this test. Therefore, the study conclusions may not be generalizable to the entire MSM population.

## Comparison with other studies

There are 2 main indicators for promoting HIV tests recommended by WHO: one is HIV testing frequency, and the other is HIV testing coverage. To our knowledge, there were only 3

prior RCTs exploring the effectiveness of HIVST kits on promoting HIV testing frequency among MSM, all of which were carried out in high-income countries [21–23]. These results cannot be directly generalized to low- and middle-income countries and areas, because of differences in socioeconomic backgrounds and disparities in public health services. WHO recommends that each country should explore culturally adaptive strategies to promote HIVST [25]. Thus, our study provides empirical evidence on the effectiveness of HIVST among MSM in China, reflecting the Chinese economic and cultural context.

In addition, WHO advocates enhancing PHT as a key approach to enlarging HIV testing coverage [25,27]. Previous research provided some evidence on the use of HIVST kits among male partners distributed by pregnant women or female sex workers [28,37]; however, these findings could not be directly applied to MSM. Our study showed that provision of free HIVST kits could more than double the potential capacity for MSM to have PHT on average. The coverage of SBHT services varies from developed countries to resource-limited countries because of different socioeconomic levels [31]. As a complementary approach to SBHT, distribution of free HIVST kits could help to reduce public health disparities and promote HIV testing coverage among MSM and their sexual partners.

A great concern was whether the implementation of HIVST kits would replace SBHT. In this study, provision of HIVST kits did not reduce the number of SBHTs. Also, distribution of HIVST kits from MSM to their sexual partners did not diminish the number of site-based PHT. HIVST supplements SBHT rather than replacing it [22,26]. Another serious concern regarding the use of HIVST kits put forward by public health scholars and CDC officers is whether MSM with positive results identified by HIVST kits would confirm their results at the CDC [38], because failure to follow-up with professional intervention and consultation could result in delayed treatment, psychological problems, or suicide. In this study, all HIV-positive MSM identified by HIVST kits were linked to HIV confirmatory laboratory tests and received ART in CDC clinics. Instructions on how to cope with positive self-testing results attached to HIVST kits, and 24-hour consultation services for positive testing results provided to participants in our study, may have contributed to the 100% linkage to care.

## Conclusion and further research

As a complementary testing approach, providing free HIVST kits could increase HIV testing frequency and enlarge HIV testing coverage among MSM and their sexual partners, and facilitate implementation of the guidelines on HIVST in China, so as to achieve the first goal of the 90-90-90 global target. However, participants recruited in this study were mainly from urban areas, whose results may not be directly generalized into rural areas. Effectiveness of HIVST on HIV testing frequency among MSM and PHT in rural areas needs to be explored.

## Supporting information

**S1 CONSORT 2010 Checklist. CONSORT, Consolidated Standards of Reporting Trials.** (DOC)

## Acknowledgments

We thank all participants for taking part in this study and the workers in Zuo An Cai Hong (the gay-friendly CBO) for their assistance in the process of recruitment. Dr. Shimin Zheng from East Tennessee State University provided much helpful consulting on statistical analysis.

## Author Contributions

**Conceptualization:** Deborah Koniak-Griffin, Lloyd A. Goldsamt, Mary-Lynn Brecht, Xianhong Li.

**Data curation:** Ci Zhang, Honghong Wang, Xianhong Li.

**Formal analysis:** Ci Zhang, Han-Zhu Qian, Xianhong Li.

**Funding acquisition:** Ci Zhang, Deborah Koniak-Griffin, Xianhong Li.

**Investigation:** Ci Zhang, Honghong Wang, Xianhong Li.

**Project administration:** Xianhong Li.

**Supervision:** Xianhong Li.

**Validation:** Ci Zhang, Han-Zhu Qian, Xianhong Li.

**Writing – original draft:** Ci Zhang, Han-Zhu Qian.

**Writing – review & editing:** Ci Zhang, Deborah Koniak-Griffin, Han-Zhu Qian, Lloyd A. Goldsamt, Honghong Wang, Mary-Lynn Brecht, Xianhong Li.

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
