## [Editor Report · Decision Letter 0]

17 Mar 2020

Dear Dr Li, 

Thank you for submitting your manuscript entitled "Impact of providing free HIV self-testing kits on frequency of testing among men who have sex with men and their sexual partners in China: a randomized controlled trial" for consideration by PLOS Medicine.

Your manuscript has now been evaluated by the PLOS Medicine editorial staff and I am writing to let you know that we would like to send your submission out for external assessment.

Kind regards,

Richard Turner, PhD

Senior editor, PLOS Medicine

rturner@plos.org

---

## [Decision Letter · Decision Letter 1]

25 May 2020

Dear Dr. Li,

Thank you very much for submitting your manuscript "Impact of providing free HIV self-testing kits on frequency of testing among men who have sex with men and their sexual partners in China: a randomized controlled trial" (PMEDICINE-D-20-00878R1) for consideration at PLOS Medicine. 

Your paper was evaluated by the editors and sent to independent reviewers, including a statistical reviewer. The reviews are appended at the bottom of this email and any accompanying reviewer attachments can be seen via the link below:

[LINK]

In light of these reviews, we will not be able to accept the manuscript for publication in the journal in its current form, but we would like to invite you to submit a revised version that addresses the reviewers' and editors' comments fully. You will appreciate that we cannot make a decision about publication until we have seen the revised manuscript and your response, and we expect to seek re-review by one or more of the reviewers. 

We hope to receive your revised manuscript by Jun 15 2020 11:59PM. Please email us (plosmedicine@plos.org) if you have any questions or concerns.

Please let me know if you have any questions. Otherwise, we look forward to receiving your revised manuscript in due course. 

Sincerely,

Richard Turner, PhD

rturner@plos.org

We ask you to make the study's primary outcome clear in the abstract, and quote the result for this outcome. Please also list the relevant secondary outcomes.

In your abstract, please quote 95% CI alongside p values. 

Throughout the paper, exact p values should be quoted unless p<0.001.

At line 50, and at an appropriate point in the main text, please quote the exact dates of the start and end of recruitment. 

If your study was registered after recruitment had begun, please explain the reason(s) for retrospective registration. 

Please add summary demographic details for study participants to your abstract.

Please quote the numbers of HIV infections in the abstract.

Please add a new final sentence to the "methods and findings" subsection of the abstract, quoting 2-3 of the study's main limitations. 

Please begin the sentence at line 61 with "In this study, we found that ..." or similar. 

Please refer to both the attached CONSORT checklist and protocol in the methods section of your main text. 

Please reformat the reference call-out at line 300. 

Please remove the baseline comparisons from table 1.

Please avoid claims of "the first", such as at line 423, and where necessary add "to our knowledge" or similar.

Please revisit your reference list to ensure that citations meet journal format. Italics should be converted into plain text. 

Please remove the competing interest statement from the end of the ms (this information will appear in the metadata (via the submission form) in the event of publication. 

Likewise, please remove the data sharing information at line 526. We note that this information appears inconsistent with that in the submission form, where you state that all relevant data are contained in the ms. PLOS' data sharing policy is that data should be presented in the ms where possible, and where this is not possible should be accessible via a public repository, for example; if a contact person is provided, this should not be an author of the article. 

Please adapt the attached CONSORT checklist so that individual items are referred to not by page or line numbers (which generally change in the event of publication) but by section (e.g., "Methods") and paragraph numbers. 

Comments from the reviewers:

Reviewer #1: 

[See attachment]

Michael Dewey

*** Reviewer #2: 

This is a well written, interesting RCT of HIVST in three cities in China. The manuscript will make a significant contribution and the authors are commended for their excellent work on an important topic. I have some comments which will assist with clarity and also are important for the interpretation of findings and the potential inclusion of this work into systematic reviews, which is inevitable given the subject matter. 

Important to note I am not a statistician and one should be consulted on the rigour and appropriateness of the analyses. 

Abstract

Clear, concise. Sets out study, main findings with clear conclusions. 

On line 46 there should be a space between '1' and 'year'

Introduction

Mentions the first 90 in the 90-90-90 targets, it is worth explaining what those targets are for those who do not know. 

Line 109 Would change 'homosexual transmission' to transmission between MSM

Line 118 - I believe the evidence is conclusive that most transmissions are from those who are not yet diagnosed. Strengthen the language here, it strengthens the argument for this paper. 

Line 121 - there are three relevant studies here, Rodger et al 2019 and 2018 

and 

Bavinton BR, Pinto AN, Phanuphak N, Grinsztejn B, Prestage GP, Zablotska-Manos IB, Jin F, Fairley CK, Moore R, Roth N, Bloch M. Viral suppression and HIV transmission in serodiscordant male couples: an international, prospective, observational, cohort study. The lancet HIV. 2018 Aug 1;5(8):e438-47.

Line 127 Key reference for values and preferences is Figueroa et al 2015. Values and preferences on HIV self-testing for key populations. 

Line 132 There are in fact two other relevant RCTs in China which should be mentioned

Tang W, Han L, Best J, Zhang Y, Mollan K, Kim J, Liu F, Hudgens M, Bayus B, Terris-Prestholt F, Galler S. Crowdsourcing HIV test promotion videos: a noninferiority randomized controlled trial in China. Clinical infectious diseases. 2016 Jun 1;62(11):1436-42. 

And 

Zhu X, Zhang W, Operario D, Zhao Y, Shi A, Zhang Z, Gao P, Perez A, Wang J, Zaller N, Yang C. Effects of a mobile health intervention to promote HIV self-testing with MSM in China: a randomized controlled trial. AIDS and behavior. 2019 Nov 1;23(11):3129-39.

Line 148 - I would replaced 'should' with 'may'. 

Methods

Line 161 - says randomised clinical trial where in other sections it says randomised controlled trial. The latter is accurate. 

Line 194 - make it absolutely explicit whether the person enrolling the participants knew what the randomisation outcome would be for the next participant. 

Line 224 - more information is needed about this kit. Would be worth including a figure with a photo and explicitly describe if it was adapted by the manufacturer for self-use. Also mention what (if any) additional modifications the study team made.

Line 233 Outcomes

On my reading of this subsection it was not clear how these were being captured from sexual partners of participants. It is really important to clarify this as it will impact on quality assessment for systematic reviewers. 

252 - make it clear in this section which (if any) outcomes were validated and how this was done. This is essential information for systematic reviewers. 

295 - patient and public involvement. On reading this is seems to be this is formative research rather than PPI. Would be worth clarifying what the process was here and what exactly was done, what questions were asked and what modifications were made in response to this. Important to demonstrate the rigour of the process. 

Results

These look good and sensible to me. Worth having checked by statistician. 

Discussions

Clear, appropriate and cautious interpretation of findings. It is important to situate these findings within the context of the other 3 RCTs in China (two of which I have provided references for earlier in this review). 

Useful discussion of limitations

Well done on excellent work. 

*** Reviewer #3: 

I appreciate the opportunity to review this article. It is an important study that addresses an interesting topic for HIV testing in a developing country setting. Overall, its RCT is very well designed, and most of the study procedure and analysis has been carefully thought through. The study findings are insightful for improving HIV testing among MSM in a developing country setting, that is, participants who are in the intervention arm, receiving HIVST kits, are being tested much more regularly and willing than the control arm. This provides important evidence for the advantages of home-based testing, which much is more approachable and user-friendly for an opaque community such as MSM. The findings that sexual partners of the participating MSM are also testing more frequently with higher coverage are also useful. This potentially outlines a practical paradigm for HIV testing in the future. Because in a society where gay relationship remains a societal taboo, enabling MSM to be self-tested at home certainly provide a useful alternative for HIV prevention and care. 

I do have several major comments related to the analysis for sexual partners of the participating MSM. 

1. The definition of sexual partners, the study did not describe the type of sexual partners, whether they are regular, casual and commercial. Giving that most MSM does have more than one partners, can a participating MSM give HIVST kits to more than one sexual partners? Based on the information in Table 3, it says nine participating MSM did not have sexual partners during 12 months of following, that is, 216 - 207 = 9. Does it mean each of the 207 MSM has exactly one sexual partner during the 12 months of follow up? 

2. How was the testing coverage and frequency of the sexual partners of the participating MSM measured, as in Table 3? Did the study reach the sexual partners directly or through the participating MSM to obtain the testing information? Were the test results of the sexual partners sent to the study administration in the same way as the participating MSM (that is, via WeChat, but they are not considered as study participants)? Do these sexual partners also conduct the three months questionnaires as the participating MSM? 

3. Besides, I am a bit confused. In both Table 1 and Table 3, the total number of HIV tests is 2? Did you mean average? In table 3, If 67.1% received ≥1 test in 12 months, does it mean 32.9% sexual partners did not receive the test? It would be better if this information is laid out more explicitly. 

There are a number of minor issues.

4. Line 179, what is the implication of 'above 35 on the Brief Psychiatric Rating Scale.'

5. Line 272-274, please identify the proportion of missing data and which indicators are missing. 

6. Line 278, the study seems to separate non-recent testers and recent testers initially and then later combines them. Then what is the point of separating them in the first place? 

7. Line 328, the flowchart, does mean the participants are being tested at the same time as the questionnaire? 

8. Line 342, Table 2, Sexual Orientation, it seems to miss a chi-2 value?

9. Line 418, although this approach has been demonstrated to effective, it will be largely based on the social network of MSM (one has to give the testing kit to his partner). I am not sure how it claims to be 'easily scaled-up through the extensive Chinese public health networks'? Not sure how the government network may facilitate the scale-up.

***

[LINK]

---

## [Decision Letter · Decision Letter 2]

23 Jul 2020

Dear Dr. Li,

Thank you very much for submitting your revised manuscript "Impact of providing free HIV self-testing kits on frequency of testing among men who have sex with men and their sexual partners in China: a randomized controlled trial" (PMEDICINE-D-20-00878R2) for consideration at PLOS Medicine. 

Your paper was re-seen by our reviewers, including a statistical reviewer. The reviews are appended at the bottom of this email and any accompanying reviewer attachments can be seen via the link below:

[LINK]

In light of these reviews, we will again be unable to accept the manuscript for publication in the journal in its current form, but we would like to invite you to submit a further revised version that addresses the reviewers' and editors' comments fully. You will recognize that we cannot make a decision about publication until we have seen the revised manuscript and your response, and we expect to seek re-review. 

We hope to receive your revised manuscript by Aug 13 2020 11:59PM. Please email us (plosmedicine@plos.org) if you have any questions or concerns.

Please let me know if you have any questions. Otherwise, we look forward to receiving your revised manuscript in due course. 

Sincerely,

Richard Turner, PhD

rturner@plos.org

At lines 36 and 83, we suggest substituting "growing" for "increasing". 

Please mention in the abstract that no adverse events were reported. 

Please quote one further study limitation around line 70. 

Please remove the information on funding at the end of the main text - this information will appear in the article metadata in the event of publication, via entries in the submission form. 

Comments from the reviewers:

*** Reviewer #1: 

[See attachment]

Michael Dewey

*** Reviewer #2: 

Well done on a great revision- I'm largely satisfied this this manuscript. I have one minor comment which I feel should be addressed before publication

1) it is great to see a clearer description of the formative research and the findings. I would note that although you conducted a qualitative study, you have reported these as quantitative outcomes. It would be more useful to have a very brief description of the themes which emerged in interviews (e.g. reasons why HIVST could increase uptake, why MSM preferred finger stick tests) and a description of how these findings were used in protocol design. If indeed this was a survey study rather than an interview study then better to change the language to reflect that.

*** Reviewer #3: 

I am satisfied with the edits the authors made, no further comments from me.

***

[LINK]

---

## [Editor Report · Decision Letter 3]

13 Aug 2020

Dear Dr. Li,

Thank you very much for re-submitting your manuscript "Impact of providing free HIV self-testing kits on frequency of testing among men who have sex with men and their sexual partners in China: a randomized controlled trial" (PMEDICINE-D-20-00878R3) for consideration at PLOS Medicine.

I have discussed the paper with editorial colleagues and our academic editor and I am pleased to tell you that, provided the remaining editorial and production issues are dealt with, we expect to be able to accept the paper for publication in the journal.

[LINK]

Please let me know if you have any questions. Otherwise, we look forward to receiving the revised manuscript shortly. 

Sincerely,

Richard Turner, PhD

rturner@plos.org

Requests from Editors:

Please revisit your data statement, with PLOS' data policy (https://journals.plos.org/plosmedicine/s/data-availability) in mind. 

First, please ensure that the data statements are consistent (currently: "all data are fully available without restriction"/"... [data] cannot be made publicly available").

Second, please substitute a non-author contact for readers wishing to inquire about data access. 

At line 47 in the abstract, you quote a p value for the difference between the number of HIV tests per participant in intervention and control arms. Following CONSORT, we usually expect to see an effect size quoted for the primary endpoint, with associated 95% CI and a p value. Are you able to provide an effect size here and in the results section?

At line 48, please make that "This difference was mainly due to the difference in HIVST between ..." (and also amend the text in a similar way at line 53). 

At line 56, please amend the text to "... intervention participants was 2.1 times greater than that of control participants ..."; and at line 58 (and line 433) "... 1.55 times more likely".

At line 59, we suggest amending the text to: "During the study period, 3 participants in the intervention arm and none in the control arm tested HIV positive, and 8 sexual partners of intervention arm participants also tested positive. No other adverse events were reported.". 

Except at the start of sentences, in the text numbers should be quoted as "3" rather than "three". 

At line 78, please make that "... shown to be effective". 

At line 92, please make that "... controlled trial". 

At line 98, please remove "increasingly expanding". 

At line 168, please make that "During development of the study protocol ...". 

At line 336, please make that "Oral consent for ...".

At line 380, please make that "after adjustment for ...".

PLOS' data policy does not, I'm afraid, permit "data not shown", and so at line 386 we ask you to include the relevant data either in table 3 or in a supplementary table. 

Around line 458, assuming our understanding is correct, please adapt the text to make it clear that the 3 HIV seroconversions in participants are known to have occurred during the study period, but that sexual partners' HIV status was not evaluated at baseline. 

Incidentally, at 457 where you note "reported by participants", we would suggest some rewording as we think that the number includes non-participants (i.e., participants' sexual partners). 

In table 2 and any other instances in the paper, please make that "Odds ratio". 

As the study protocol is published, we suggest removing the attached file. 

***

---

## [Editor Report · Decision Letter 4]

31 Aug 2020

Dear Dr. Li, 

On behalf of my colleagues and the academic editor, Dr. T Charles Witzel, I am delighted to inform you that your manuscript entitled "Impact of providing free HIV self-testing kits on frequency of testing among men who have sex with men and their sexual partners in China: a randomized controlled trial" (PMEDICINE-D-20-00878R4) has been accepted for publication in PLOS Medicine. 

PRODUCTION PROCESS

PRESS

PROFILE INFORMATION

Thank you again for submitting the manuscript to PLOS Medicine. We look forward to publishing it. 

Best wishes, 

Richard Turner, PhD

Senior Editor 

PLOS Medicine

plosmedicine.org